# Reduced Carbonation, Sulfate and Chloride Ingress Due to the Substitution of Cement by 10% Non-Precalcined Bentonite

**DOI:** 10.3390/ma14051300

**Published:** 2021-03-08

**Authors:** Carmen Andrade, Ana Martínez-Serrano, Miguel Ángel Sanjuán, José Antonio Tenorio Ríos

**Affiliations:** 1International Center of Numerical Methods in Engineering (CIMNE)-UPC, 28010 Madrid, Spain; candrade@cimne.upc.edu; 2Institute of Construction Sciencies “Eduardo Torroja”-CSIC, 28033 Madrid, Spain; ana.martinez@ietcc.csic.es; 3Institute of Cement and Its Applications (IECA), 28003 Madrid, Spain; masanjuan@ieca.es

**Keywords:** cement, bentonite, durability, clays

## Abstract

The Portland cement industry is presently deemed to account for around 7.4% of the carbon dioxide emitted annually worldwide. Clinker production is being reduced worldwide in response to the need to drastically lower greenhouse gas emissions. The trend began in the nineteen seventies with the advent of mineral additions to replace clinker. Blast furnace slag and fly ash, industrial by-products that were being stockpiled in waste heaps at the time, have not commonly been included in cements. Supply of these additions is no longer guaranteed, however, due to restrained activity in the source industries for the same reasons as in clinker production. The search is consequently on for other additions that may lower pollutant gas emissions without altering cement performance. In this study, bentonite, a very common clay, was used as such an addition directly, with no need for precalcination, a still novel approach that has been scantly explored to date for reinforced structural concrete with structural applications. The results of the mechanical strength and chemical resistance (to sulfates, carbonation and chlorides) tests conducted are promising. The carbonation findings proved to be of particular interest, for that is the area where cement with mineral additions tends to be least effective. In the bentonite-bearing material analysed here, however, carbonation resistance was found to be as low as or lower than that observed in plain Portland cement.

## 1. Introduction

The inordinate rise in the presence of greenhouse gases in the atmosphere is creating a pressing need to lower CO_2_ emissions by, among other methods, reducing the proportion of clinker in cement [1,2,3]. The Portland cement industry is presently deemed to account for around 7.4% of the close to 2.9 Gt of carbon dioxide emitted annually worldwide (value for 2016) [4]. In light of such facts, the industry is assessing the measures that could be taken to ensure zero emissions by 2050. Under review are the processes involved in clinker, cement and concrete production, construction procedures and cement-based material carbonation during and after service life [5]. As the mitigation technologies presently in place are believed to be insufficient to hit the net zero carbon target by 2050, innovative measures are called for, including carbon dioxide capture, utilization and storage (CCUS) [6] and flameless mineral calcination systems. One new proposal for the latter, the tube-in-tube helical method, features use in concentrated solar power plants [7].

In the past, clinker content has been replaced with industrial waste such as fly ash, slag or silica fume with no adverse effect on concrete mechanical performance or durability [8]. With the abatement of the likewise carbon-intensive source industries, however, the availability of those substitute mineral additions is beginning to wane. Although the contribution of coal power plant-fueled energy to total consumption declined in Spain from 20.2% in 1990 to 9.8% in 2017 thanks to the growing use of alternative energies [9], coal continued to supply 38.5% of world demand in 2018. Concerns about greenhouse gas emissions cloud the future of coal, however, for it is pivotal to the debate on energy and climate policy. A number of countries, committed to net-zero greenhouse gas (GHG) emission targets by 2050, have established an end date for coal power generation. In others, however, coal plays a key role in the supply of affordable energy [10]. That notwithstanding, coal’s share in the global power mix is expected to decline by 10% by 2050, with a concomitant downturn in the stock of fly ash. Hence, it is imperative to seek replacements for clinker in nature to broaden the spectrum of alternative materials. The study described hereunder explored the use of non-precalcined bentonite, a widely available clay, as one such alternative.

As an anionic clay present in nature, bentonite can be obtained at low cost. It is also highly water absorbent and thixotropic (gel-like when vibrated). It was first discovered in 1988 in the United States and more specifically at Fort Benton, Wyoming [11], after which it is named. Its composition consists primarily of magnesium silicate, montmorillonite and aluminium hydrate, the third in the form of colloid-sized crystallites. Each individual montmorillonite crystal, in turn, comprises an octahedral layer of aluminium sandwiched between two tetrahedral layers of silicon. It carries a negative charge associated with isomorphic substitutions, such as Al^3+^ for Mg^2+^, in the crystallite network, which is offset by exchangeable alkaline metal cations [11].

Clay use as a mineral addition in cement is nothing new, although in most cases subject to precalcination [12]. A number of studies [13,14,15] have recently reported promising results around its application to replace more conventionally used additions such as fly ash or slag. Few studies on its non-precalcined use have been found in the literature, however, for that approach has consistently posed rheological problems [16,17,18,19,20,21,22,23,24,25,26]. That would explain why many building codes limit the presence of clay materials in aggregates and the much more common use of these materials in foundations and soil stabilisation than for structural applications [27,28,29]. Nonetheless, today’s admixtures afford fresh concrete properties impossible to attain in the past, and modern laboratory techniques now in place can substantially shorten the time needed to design an optimal mix [30,31,32,33]. Non-precalcined bentonite has seldom been used to date in lieu of more conventional mineral additions [16,17,18,19,20,21,22,23,24,25,26]. This study was therefore designed to study the hydration mechanisms involved [19,24,26], fresh concrete properties such as flowability and bleeding [18,25], bentonite reactivity [19] and concrete water permeability [17,20,22] and compressive strength [16,25]. Durability was studied in terms of the replacement’s effect on concrete resistance to freeze–thaw cycles and acid or sulfate attack [23]. Bentonite was reported to have a beneficial impact on preventing reinforcement corrosion [20] and carbonation resistance [21], although only one paper on each subject was located in the literature.

A review of the literature on the long-term performance of bentonite showed that it has exhibited excellent durability in underground works, where it has been used profusely as permanent formwork in concrete foundations often built long ago [27,28,29]. Additionally, whilst bentonite plays a non-calculated load-bearing role in such cases, none of the studies published report any long-term incompatibility between the two materials. It is likewise used in conjunction with concrete to generate impermeable slurry walls in highly radioactive waste storage facilities (designed to last for thousands of years) [34,35,36], where the caverns holding the radioactive waste are shotcreted and the encapsulated waste itself lies on a bed of bentonite. The use of such systems has given rise to research on how the alkaline nature of concrete may affect the long-term stability of bentonite clays. Such studies have verified the interaction between cement alkalinity and bentonite phases [34,35,36,37] or, equivalently, phase reactivity with clinker hydrated phases. From the standpoint of the role of clay as a clinker replacement, the findings have been initially promising thanks to the slow reactivity afforded by the high alkalinity of the pore solution, which dissolves the silicoaluminates in the bentonite. In terms of radioactive waste, such a result would be detrimental, however, if it affected the stability of the shotcrete/bentonite interface, given the many thousands of years they are intended to be in contact.

Standardised active additions such as fly ash, natural pozzolans, slag and silica fume, in turn, are known to effectively inhibit chloride and sulfate ingress [38,39], enhancing durability, although their presence in concrete impacts carbonation resistance adversely [13,16]. That is significant, for any decline in carbonate resistance is a primary long-term concern because it favours reinforcement corrosion [40,41,42] and the associated economic loss. The importance of seeking additions that either favour or at least are not detrimental to concrete durability cannot, therefore, be overstated.

Eluding the extra cost and additional handling involved in precalcining clays at temperatures of up to 1000 °C would carry obvious advantages. The present study consequently aims to explore the physical-mechanical properties and durability of concrete prepared with a non-precalcined bentonite as a substitute for clinker at different (wt/wt) replacement ratios.

This study sought to determine how replacing up to 30% clinker with non-precalcined bentonite may affect mortar mechanical properties and how carbonation depth and chloride and sulfate diffusion may be impacted by the presence of 10% of the clay with different types of binders bearing mineral additions. The findings are highly promising, particularly as regards carbonation, the weak point observed in other mineral additions. In the tests conducted, carbonation resistance either remained essentially unchanged or improved in the mortar prepared with the blended cement relative to the reference material. Plain Portland cement or cement bearing standardised additions was used throughout [43].

## 2. Materials and Methods

### 2.1. Materials

All the cements listed in Table 1 were used in the carbonation tests, whereas cement CEM I 52.5 R-SR 3 served as the basis for the mechanical strength and sulfate and chloride diffusion trials.

A commercial bentonite (Mapeproof Seal), distributed by Mapei for purposes other than those studied here, was used to ensure consistent composition and particle size distribution throughout. According to the vendor’s specifications sheet, the material contained over 95 wt % montmorillonite.

The particle size distribution and volume density curves were found by analysing bentonite powder on a Mastersizer 3000 laser diffractometer diffractor (Malvern Panalytical, Madrid, Spain) (Figure 1). Ninety per cent of the particles were <86 µm, whilst most lay within the 10 to 15 µm range (see the volume density curve). Such greater fineness than observed for the cement was initially deemed suitable, although optimisable. Addition fineness plays a significant role in the strength and rheology of composite cements, for the distribution curves for those materials complement the curve for the cement itself. This parameter must consequently be analyzed in depth in future research [43,44].

Another factor of particular interest in bentonite materials, which lies outside the scope of the present study, is water demand, affected not only by fineness but also by its sodium content [8].

In all the pastes and mortars prepared, bentonite replaced the corresponding binder content, and the water cement ratio was calculated as cement plus bentonite: w/cm.

### 2.2. Specimen Types

Different types of specimens were prepared, depending on the test.

-For mechanical strength and carbonation resistance testing, 10 × 10 × 60 mm cement paste specimens bearing 10%, 20% or 30% bentonite additions were prepared at a water/cement ratio of 0.5. They were cured in a climatic chamber at 90% relative humidity, first in the moulds for 24 h and after removal for 28 d prior to testing.-For chloride diffusions, the 70 cubic mm cement mortar specimens used were prepared with a water/cement ratio of 0.5. They were cured in a climatic chamber at 90% relative humidity, first in the moulds for 24 h and after removal for 28 d prior to application of an electric current to test for chloride diffusion.

### 2.3. Test Methods

#### 2.3.1. X-ray Diffraction

The mineralogical composition of the cement paste ground and sieved to 45 µm [40] was determined on a Bruker AXS DB Advance X-ray diffractor (Bruker, Madrid, Spain) configured without a monochromator, fitted with a 3 kW (Cu Kα1.2) copper anode X-ray source and a wolfram cathode. A 30 mA current was applied to the X-ray tube at a voltage of 40 kV. A 0.5 mm fixed divergence slit was used. The instrument was also fitted with a 2.5 rad primary Soller slit and a Lynx-eye X-ray super-speed detector diffractor (Hamamatsu, Hamamatsu, Spain) with a 3 mm anti-scatter slit, a 2.50 rad secondary Soller slit and a 0.5% Ni-K beta filter. The specific reflection peak used was 2θ = 35°.

#### 2.3.2. Twenty-Eight Day Flexural and Compressive Strength

Testing for flexural strength [42] consisted in bending the prismatic specimens by applying a force perpendicular to their longitudinal axis, on a Netsch test frame specifically designed for small specimens.

The test was deemed valid only when the specimen failed across the middle.

The two halves of the specimens resulting from the flexural test were subsequently used for compression testing.

Compressive strength was found by exposing the specimens to two axial forces with equal modulus and orientation but coursing in opposite and convergent directions, on an Ibertest Autotest 200/10-SW test frame [45].

#### 2.3.3. Carbonation in Natural Environments

The cements tested and their chemical compositions are given in Table 2. The 10 × 10 × 60 mm specimens were exposed to natural carbonation at the atmospheric CO_2_ pressure prevailing in the city of Madrid, in an indoor laboratory environment and two outdoor environments, one sheltered and the other unsheltered from rainfall, i.e., environments with varying relative humidity and temperatures (Figure 2).

Carbonation depth as found with phenolphthalein, an acid-base indicator, was re-corded for the 3-month and 6-month specimens, depicted in the three environments in Figure 2.

#### 2.3.4. Sulfate Resistance: The Koch–Steinegger Method

Cement paste resistance to sulfate ions was tested on 10 × 10 × 60 mm specimens further to the Koch–Steinegger method, based on comparing flexural strength in such specimens soaked for 56 d in an aggressive solution (here, sodium sulfate at a concentration of 4.4 g/L) to the strength of analogous specimens soaked in water, likewise for 56 d (Figure 3). All the specimens had been cured in a humidity chamber for 28 d prior to testing.

#### 2.3.5. Accelerated Chloride Ingress

The accelerated chloride diffusion test described in Spanish standard UNE 83992-2 EX [36] was conducted on 70 cubic mm mortar specimens, each bearing an embedded steel bar. Performance by the samples with 10% bentonite was compared to the results observed for reference CEM I 42.5SR specimens of the same dimensions.

The test consisted in connecting specimens made with different types of mortar to an electrical current that accelerated chloride ion diffusion (migration) across the matrix toward the bar (see setup in Figure 4). The steel was assumed to begin to corrode when surface contact with the chlorides was electrochemically detected. That initial corrosion time and the amount of chloride on the bar surface were the parameters used to calculate the diffusion coefficient.

## 3. Results

### 3.1. Flexural and Comprenssive Strength

Additions should not alter, except to improve, mix mechanical performance. As Figure 5a shows, replacing 10% or 20% of cement CEM I 52.5R–SR 3 with bentonite raised 28 d flexural strength relative to the reference cement except at a replacement ratio of 30%. Adding 30% bentonite yielded lower compressive strength than in the reference and in the materials with 10% or 20% replacement.

While unaffected by bentonite at a replacement ratio of 10% (Figure 5b), compressive strength declined at ratios of 20% or 30%. Those findings informed the decision to use only the 10% bentonite in all the subsequent tests as the most conservative option.

### 3.2. X-ray Diffraction-Based Characterisation

The possible reactivity and stability of bentonite-bearing cement paste were also explored. The diffractograms for 28 d pastes bearing 10%, 20% and 30% bentonite are reproduced in Figure 6, whilst the relative content (counts, in per cent) of the various phases is graphed in Figure 7.

The reflections attributable to bentonite (montmorillonite and quartz) rose in intensity between 10% and 20% replacement, although no such rise was visible between 20% and 30% [44].

Portlandite content was similar to the reference in the former two mixes, but declined significantly at 30% replacement. According to [34], that decline might be explained by the formation of a calcium zeolite in the interaction between bentonite and the pore solution in the hydrated cement; confirmation of such a premise lies outside the scope of the present study.

Inasmuch as the carbonate phases would have been generated by carbonation occurring during the test, for the time being the rise in intensity with bentonite content need not be attributed to that higher proportion of the clay.

As ettringite content, in turn, followed neither an upward nor a downward pattern, its presence would be due to the cement and therefore may be deemed unaffected by the addition.

### 3.3. Sulfate Attack

The results for this test are deemed acceptable when the strength of the blended cement is greater than 70% of the value recorded for the control soaked in distilled water. Further to the flexural and compressive strengths of the reference specimen soaked in water and the specimens bearing 10% bentonite graphed in Figure 8, strength was higher in both the reference and in the specimen bearing the bentonite presence when soaked in the sulfate solution than when soaked in water. That rise in strength was attributed to the higher degree of hydration resulting from the difference in specimen ages: 56 d rather than 28 d. The values observed for the reference paste and the paste prepared with the blended cement were very similar. On the grounds of those data, the presence of bentonite may be deemed to have had no effect on cement resistance to sulfate attack.

### 3.4. Chloride Resistance Test

This test aimed to determine the effects of the bentonite addition on chloride transport in the mortar matrix and the chloride ion threshold at which the reinforcing steel began to corrode. Figure 9 shows corrosion potential and corrosion rate from time 0 until an abrupt change in tendency in the respective curves denoted the onset of reinforcement depassivation.

The chloride diffusion coefficients calculated from the penetration depth of the colorimetric front depicted in Figure 10 are listed in Table 3. Much smaller values were observed for the mortar bearing 10% bentonite. Rather than penetration depth per se (the red line in Figure 10), the decline reflects differences in test times, for depassivation occurred in the reference earlier than in the bentonite-bearing material. In other words, it took much longer to reach the penetration shown in the figures in the bentonite-bearing than in the reference specimens, denoting higher electrical resistivity in the former.

Further to the chloride content data given in Table 4, surface concentration was higher, whilst the chloride threshold was, inversely, lower, in the presence of bentonite. This allows one to confirm that it is the transport phase expressed in the diffusion coefficient which controls the better behaviour of the mortar with bentonite.

### 3.5. Natural Carbonation

The cumulative rainfall recorded in Madrid in the 6 months the specimens were exposed to outdoor conditions was 160 L/m^2^, whilst the mean temperature was on the order of 30 °C. The photographs in Table 5 depict the phenolphthalein staining in the specimens from which carbonation depth was deduced. Generally speaking, the shallowest depths were observed in the laboratory, intermediate penetration under outdoor sheltered conditions and the deepest in the specimens exposed to rainfall. That order of environmental aggressiveness is diametrically opposed to earlier reports. According to those data, penetration was deepest in specimens exposed to indoor environments or sheltered outdoor conditions, whilst carbonation was least intense in those exposed to rainfall, due to their higher or nearly optimal moisture content. The present findings were deemed accurate, however, for they were qualitatively identical in the 3-month and 6-month specimens. In this study, the specimens exposed for 6 months were also tested during the summer under high-temperature, low relative humidity conditions. In other seasons, with higher RH and more rain, the order may have differed. That is scantly relevant, however, for inasmuch as carbonation was intense in all the samples, the findings sufficed for the aim pursued, namely to compare the behaviour in the various cements.

Figure 11 plots the 3-month carbonation depths in the specimens bearing 10% bentonite against the respective references, and Figure 12 plots the same parameters in the 6-month samples. After 3 months, carbonation was less intense in a larger number of 10% bentonite than in reference specimens. The gap was smaller after 6 months, although in some cases carbonation was more intense in the blended cement than in the reference specimens.

As a rule, the flexural strengths (Figure 13, Figure 14, Figure 15 and Figure 16) were fairly similar in the reference and blended samples. The compressive strength values were even closer in the two types of mortars. Inasmuch as the experiment was designed for purposes of comparison, the inference drawn from these findings is that replacing 10% bentonite in the cement mix had no material effect on mortar behaviour.

## 4. Discussion

Rising to the challenge posed by the need to reduce the cement industry’s carbon footprint may involve either large-scale technological change based on research into new manufacturing methods or adopting a more direct and technologically simple approach consisting in lowering the proportion of clinker in cements without altering their essential properties [1,2,3]. Replacing clinker with non-CO_2_-emitting materials is the most immediate alternative open to the industry.

By-products such as blast furnace slag, that are in themselves cementitious, have long been deployed to reduce clinker content [1], as have acid materials (pozzolans and more recently fly ash and silica fume) that react with the calcium hydroxide released in cement hydration. The use of natural pozzolans began to decline in the wake of their depletion in some natural reserves or because of the adverse impact of quarrying on the environment. In contrast, as an industrial by-product, fly ash was much less costly, for it did not have to be mined and the enormous stockpiles of coal industry waste had to be pared down. At around the same time, the nineteen seventies oil crisis raised the price of the fuel used to manufacture clinker. That combination of factors led to a general trend to add minerals to the clinker, which, depending on the commercial and legislative conditions prevailing in any given country, were either milled directly with the clinker or added at concrete plants.

As noted in the introduction, the evidence of climate change and the gradual reduction of the stock of such by-products have driven a return to former paradigms, including the use of additions other than slag or fly ash, such as precalcined clays [12]. Even with the investment involved in precalcination, clay has become competitive due to the rising cost of emissions [14]. Indisputably, however, the initial cost would be even more competitive if cement performance could be ensured with no need for precalcination.

Such precalcination entails, among other consequences, the loss of the bound water in the constituent minerals present in clay [8,9,10,11,12], which is recovered during hydration. Such thermal dehydration affects clay reactivity, i.e., cement hydration kinetics, but should not in principle impact component stability, for the hydrated compounds at issue are the same as they would be if the clay were used without precalcination. That is one of the many matters in connection with the use of non-pre-dehydrated natural clays, bentonite among them, in need of more thorough research.

In light of its vast diversity and fairly widespread geographic availability, clay is just one more local raw material [17,20] deployed in cement manufacture. The very same clays used in clinker kilns might, in certain proportions, constitute compatible additions to lower milling-related CO_2_ emissions (since milling clay is less energy-intensive than grinding clinker). That reasoning informed the initiative to undertake exploratory research along those lines, part of the results of which are described hereunder.

Surprisingly, very few studies [16,17,18,19,20,21,22,23,24,25,26] on the subject were found in the literature other than reports of the widespread joint use of cement and bentonite in underground works [28,29], such as the nuclear waste storage [34,35,36,37,38], soil stabilisation or impermeable slurry wall construction [27].

To be compatible with cement and usable in concrete, additions must meet a series of short- and long-term requisites, summarised below.

-They must be inert or at least not induce expansion or degenerative reactions.-They must improve or at least not alter concrete volume stability (in terms of shrinkage and creep especially).-They must improve or at least not alter mechanical performance.-They must lengthen or at least not shorten concrete or steel durability.

This study addresses some but not all of those factors. The findings are deemed sufficiently promising to be made public, acknowledging, however, that the use of non-precalcined clays will call for considerable research, in light of their enormous variety.

The following paragraphs discuss the more or less basic features of the use of non-precalcined clays analysed here, i.e., the effect on mechanical strength, the nature of the hydration products forming and the impact of bentonites on resistance to sulfates, chloride ingress and carbonation.

Be it said from the outset in connection with flexural and compressive strength that bentonite thixotropy necessitates adjusting admixtures and a constant w/cm ratio to ensure suitable mix workability [31,32]. Such thixotropy, which has not been studied in depth, may be either a drawback or an advantage in terms of workability, depending on the intended application of the concrete (such as precasting or additive manufacturing, also known as 3D printing) at issue [43]. In this study, carboxylate admixtures [31,32] were the simplest choice to avoid mixing problems.

One of the most prominent findings of this research was the rise in flexural strength with the proportion of bentonite. In the absence of supplementary testing, no reasons for such a rise can be ventured at this time. Although compressive strength was observed to decline as the replacement ratio rose, that development was readily attributable to the concomitantly lower clinker content.

The XRD findings for the hydrated pastes revealed that at the ages studied the cement barely reacted with bentonite. Further to reports on underground structures, for the nuclear industry in particular, high cement alkalinity induces the formation of a certain proportion of calcium silicate hydrates and calcium zeolites [35,36]. At ambient temperatures, however, that reaction is apparently slow enough to deem bentonite a nearly inert substance.

Although the results of the Koch–Steinegger sulfate resistance tests might be dismissed for their failure to represent actual conditions, they are nonetheless indicative of relatively short-term anomalous and expansive reactions. Longer-term tests using different solutions would be required to confirm the present initially promising results in this regard.

The lower diffusion coefficient measured for chloride ingress, in turn, was attributable to the timing differences between the tests conducted with the reference and with bentonite [46]. As the clay retards chloride penetration significantly [20], its use as an addition would be beneficial, although further research is called for to determine the reasons for this behaviour. One possibility might be the reduction of porosity (parameter not measured here), whereas any reaction between bentonite and chlorides would be all but ruled out in light of the negative charge in the clay’s interlayers, which would accommodate cations but not anions.

The effect of bentonite on carbonation depth must be assessed in the realisation that its action supplemented the action of other additions present in the cement. In other words, at least two mineral additions were in place in the tests conducted here, accounting in some cases for a substantial fraction of the total. In the two blended CEM I cements used, carbonation was the same or even lower than when no bentonite was present. Of the other cements, the ones bearing natural pozzolans appeared to perform better than those carrying fly ash or slag. In neither case did the use of bentonite induce clearly poorer performance than already observed in those cements. However, such behaviour cannot be attributed to a reaction between the clay and carbon dioxide, for as noted above, bentonite cannot accommodate anions in its interlayers [47,48,49].

This feature, the reduction or at least non-alteration of carbonation depth, is deemed to be the most relevant finding of this study. For the opposite, lower carbonate resistance is one of the shortcomings identified in mineral additions in general. Bentonite could consequently be used to advantage instead of the 5% of inert matter or the up to 10% of limestone routinely added to clinker. It may improve one or several properties of the end product. Confirmation of the foregoing will nonetheless call for much more testing, in particular to detect possible adverse effects on shrinkage or creep.

In the context of the pursuit of a circular economy and climate change mitigation, the cement industry is undertaking new strategies to reach a net zero emissions target by 2050. One such strategy, the production and use of blended cements with a high pozzolanic material content, makes the need to find new additions the more pressing [5]. Bentonite is a well-known clay consisting mostly of montmorillonite, an aluminium phyllosilicate mineral whose microscopic (~1 µm in diameter) plate-shaped particles afford the clay a large surface area. Precalcined clay has been standardised (European standard EN 197-1:2011 [50]) under the category ‘natural calcined pozzolana (Q)’, defined as thermally treated clays, shales, sedimentary rocks or materials of volcanic origin. Inasmuch as bentonite is a clay that requires no thermal activation, it might well be classified under the designatory letter ‘Q’.

## 5. Conclusions

This article describes exploratory experimentation on some of the properties of pastes and mortars made with different proportions of bentonite. The conclusions that may be drawn from the findings include the following.
Replacing up to 20% cement with bentonite enhances flexural, and up to 10%, compressive strength.At 10%, the addition induces no change in cement crystalline hydration products or in the components of bentonite itself (>95% montmorillonite further to supplier specifications).Replacing 10% of the cement with bentonite:
a.In a sulfate solution for 56 d raises cement paste mechanical strength relative to the same materials stored in distilled water, attributed to greater age in the absence of expansive reactions;b.Lowers the chloride diffusion coefficient significantly;c.Reduces or maintains the carbonation depth observed in the reference material, deemed to be a very promising development.

New and innovative measures must be taken by the cement industry worldwide to minimise its impact on climate change. One effective approach to reaching carbon neutrality consists in using new constituents to manufacture Portland cement. Insofar as bentonite, while a clay material, calls for no thermal activation, the present authors suggest that it be classified under the European standard EN 197-1:2011 [50] heading ‘natural calcined pozzolana (Q)’.

## Figures and Tables

**Figure 1 materials-14-01300-f001:**
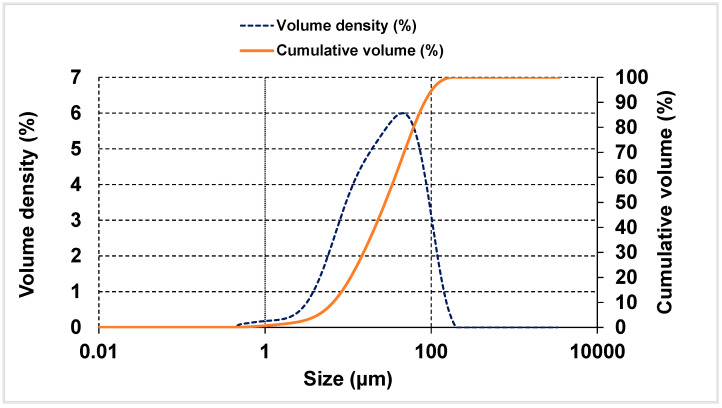
Particle size distribution and volume density plots for bentonite.

**Figure 2 materials-14-01300-f002:**
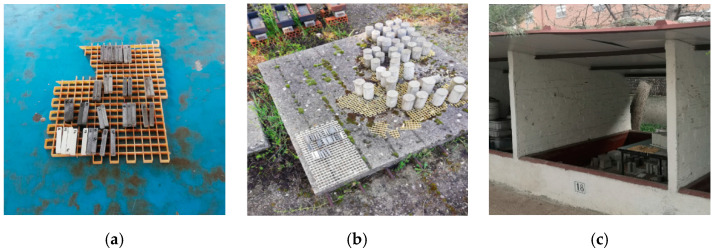
Exposure to natural carbonation: environments: (**a**) indoor (laboratory) environment, (**b**) outdoor unsheltered environment and (**c**) outdoor sheltered environment.

**Figure 3 materials-14-01300-f003:**
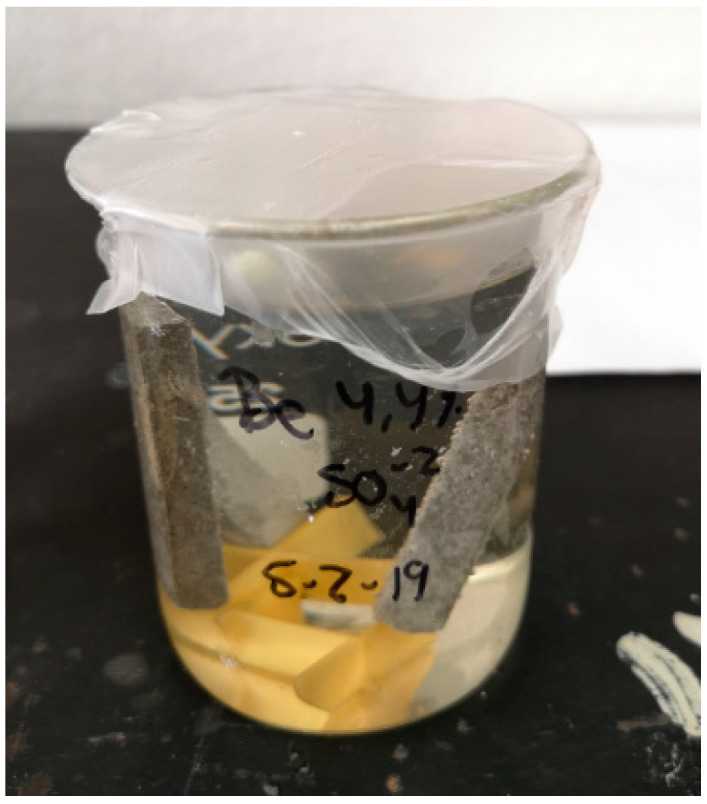
Koch–Steinegger exposure to sulfate attack.

**Figure 4 materials-14-01300-f004:**
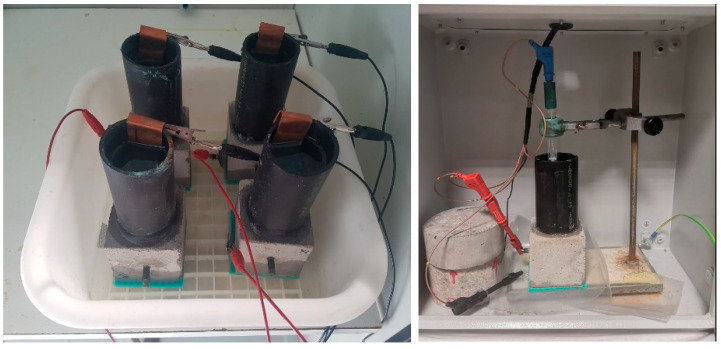
Accelerated corrosion test setup.

**Figure 5 materials-14-01300-f005:**
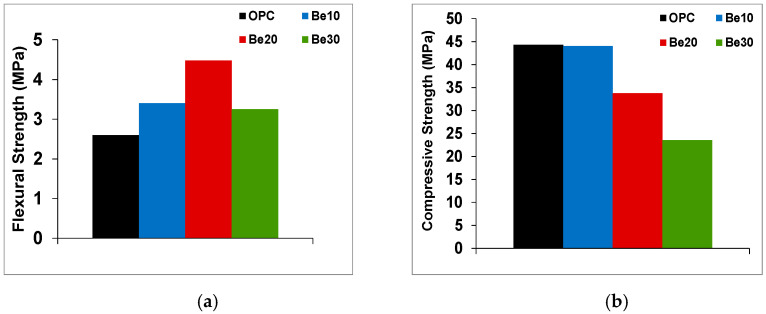
(**a**) Flexural and (**b**) compressive strength in cement pastes prepared with the reference and blended cements at replacement ratios of 10%, 20% or 30%.

**Figure 6 materials-14-01300-f006:**
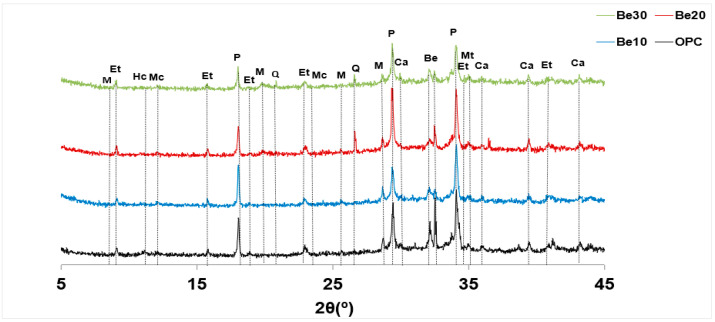
XRD patterns for unadditioned CEM I 52.5R–SR 3 and the same cement with 10%, 20% or 30% bentonite.

**Figure 7 materials-14-01300-f007:**
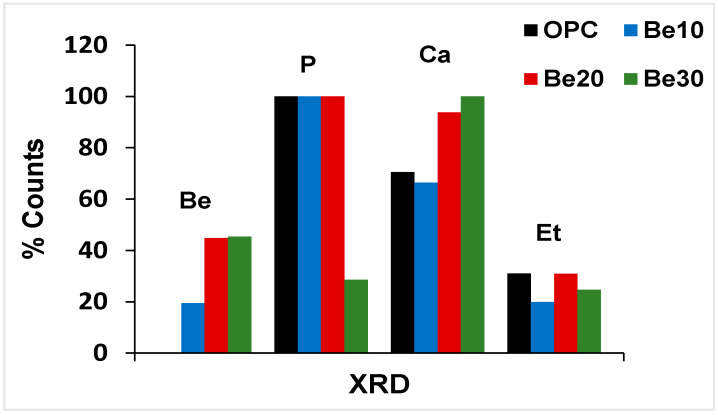
Crystalline phases identified on the XRD patterns for the reference and bentonite-bearing mixes: counts (%).

**Figure 8 materials-14-01300-f008:**
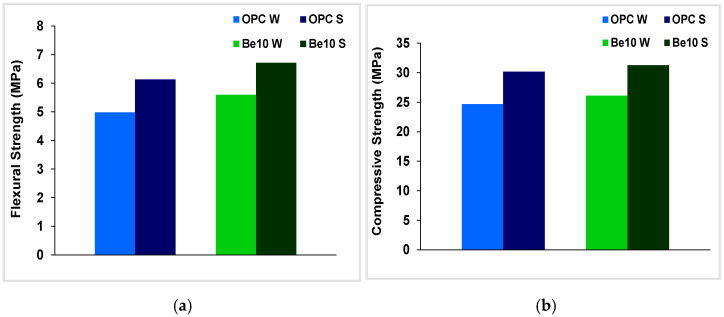
(**a**) Flexural and (**b**) compressive strength in reference specimens and specimens bearing 10% bentonite soaked in distilled water (W) or in sulfate (S).

**Figure 9 materials-14-01300-f009:**
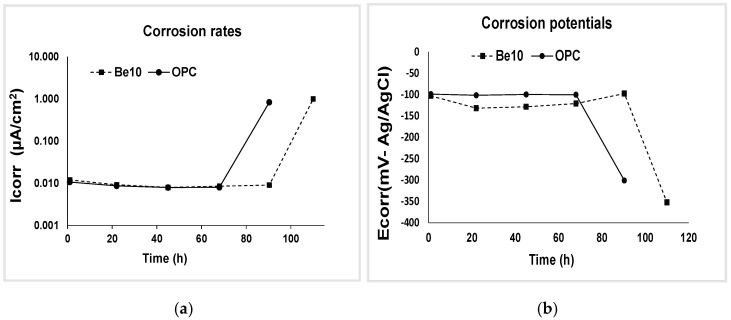
(**a**) Corrosion potential (0 to 100 h) and (**b**) corrosion rate (0 to 120 h), illustrating the abrupt change in tendency that denotes the onset of reinforcement depassivation.

**Figure 10 materials-14-01300-f010:**
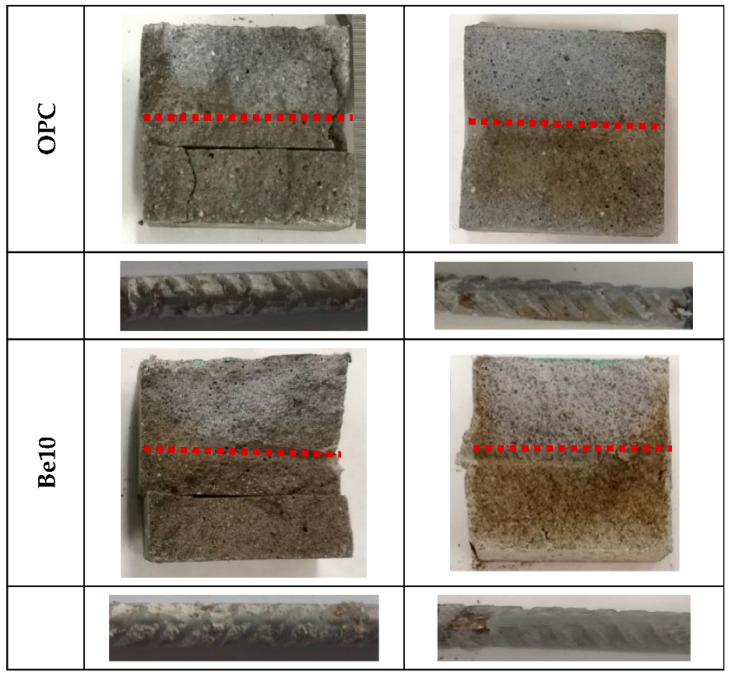
Chloride penetration front upon finalization of the experiment with the detection of the onset of corrosion.

**Figure 11 materials-14-01300-f011:**
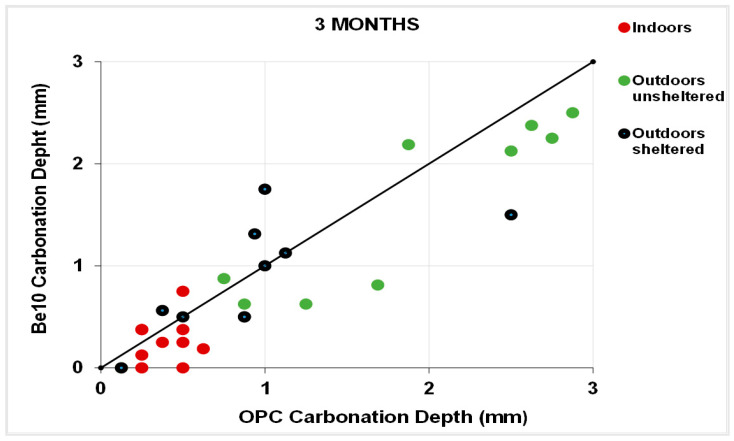
Three-month carbonation depth in reference OPC vs. 10% bentonite-bearing Be10.

**Figure 12 materials-14-01300-f012:**
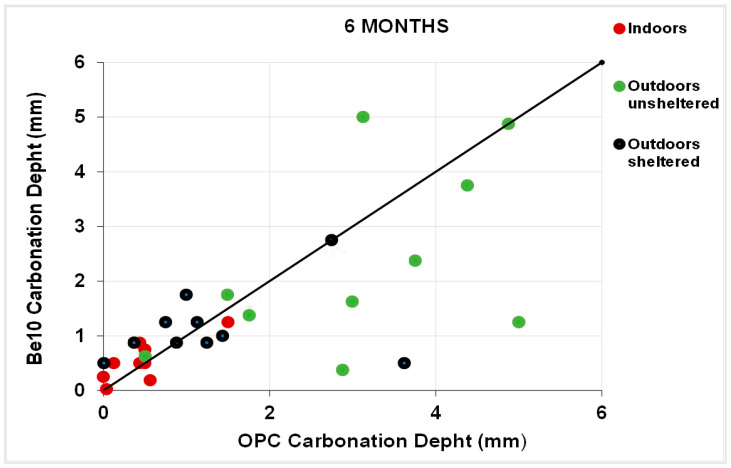
Six-month carbonation depth in reference OPC vs. 10% bentonite-bearing Be10.

**Figure 13 materials-14-01300-f013:**
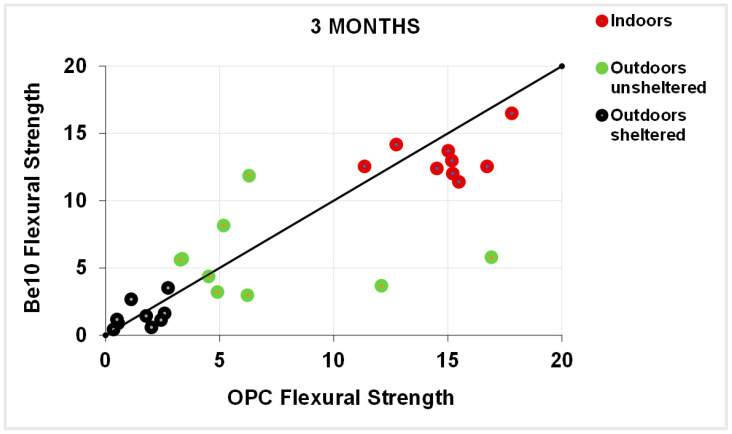
Three-month flexural strength: OPC vs. Be10.

**Figure 14 materials-14-01300-f014:**
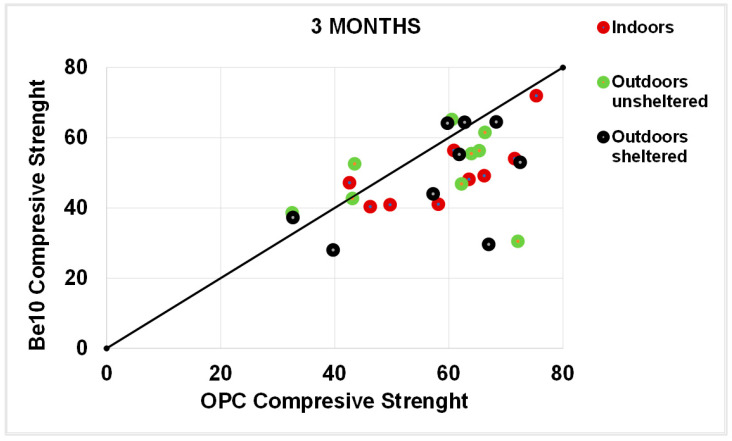
Three-month compressive strength: OPC vs. Be10.

**Figure 15 materials-14-01300-f015:**
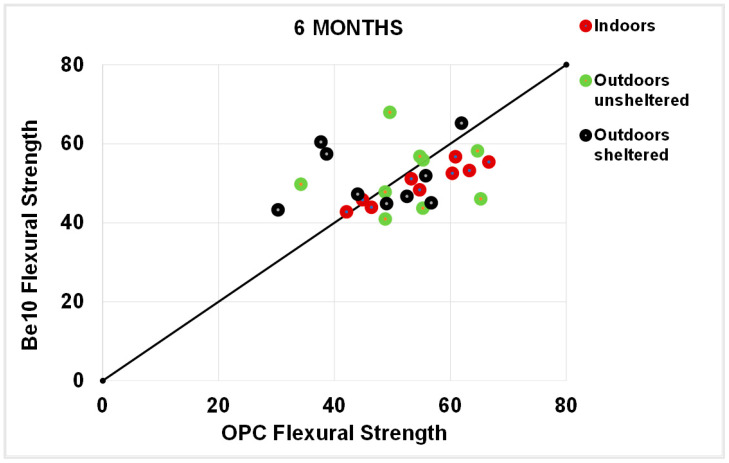
Six-month flexural strength: OPC vs. Be10.

**Figure 16 materials-14-01300-f016:**
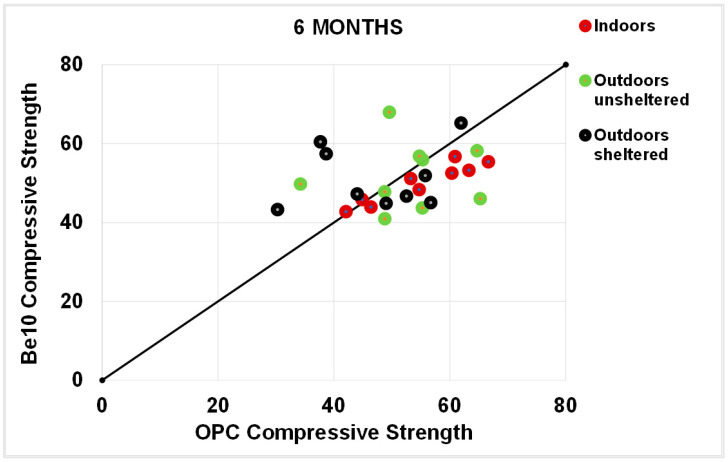
Six-month compressive strength: OPC vs. Be10.

**Table 1 materials-14-01300-t001:** Chemical composition of the cements used in the tests.

Cement	SiO_2_	Al_2_O_3_	Fe_2_O_3_	CaO	MgO	SO_3_	Na_2_O	K_2_O	LOI	IR	Cl^−^
CEM I 42.5 R	20.24	3.99	2.92	62.88	1.41	3.47	0.08	0.86	2.78	0.10	0.02
CEM I 52.5R–SR 3	21.73	3.67	4.31	66.12	1.32	3.00	0.49	0.57	1.12	0.19	0.01
CEM II/A-P 42.5 R	29.09	5.29	2.98	51.60	1.78	2.82	0.39	0.53	2.94	-	0.06
CEM II/A-P 42.5 R	28.17	6.20	3.13	52.41	1.32	3.11	0.38	0.67	2.34	--	0.05
CEM II/A-S 42.5 N	23.24	5.74	2.46	61.82	2.29	2.83	0.46	0.59	-	-	0.05
CEM II/A- V 42.5 R	23.00	6.30	3.50	58.00	1.42	3.22	0.49	0.80	2.30	2.10	0.06
CEM III/A 42.5 N	24.55	6.42	2.14	57.14	3.00	2.80	0.40	0.50	0.91	0.21	0.05
CEM IV/A (V) 42.5 R-SR	27.14	5.25	3.20	53.10	1.58	2.82	0.37	0.49	2.70	-	0.06
IV/A(P) 42.5 R/MR	28.36	4.72	3.17	52.81	2.16	2.59	0.33	0.51	2.43	-	0.04
BL II/B-LL 42.5 R	18.70	3.83	2.64	61.70	1.29	2.97	0.06	0.81	10.86	0.33	0.02

**Table 2 materials-14-01300-t002:** Composition (%) of the cements used in the tests.

Cement	K	V	L/LL	S	P	Addition
CEM I 42.5 R	95					5
CEM I 52.5R–SR 3	95					5
CEM II/A-P 42.5 R	83				11	6
CEM II/A-P 42.5 R	80				16	4
CEM II/A-S 42.5 N	83			12		5
CEM II/A-V 42.5 R	80		15			5
CEM III/A 42.5 N	59			39		2
CEM IV/A (V) 42.5 R-SR	74	23				3
IV/A(P) 42.5 R/MR	87				13	0
BL II/B-LL 42.5 R	74		26			0

**Table 3 materials-14-01300-t003:** Chloride non-steady-state non steady-state diffusion coefficient (D_ns_) in reference (OPC) specimens and samples bearing 10% bentonite (Be10).

Sample	Test Time (h)	Maximum Penetration (mm)	Mean Penetration (mm)	D_ns_ Diffusion Coefficient (cm^2^/s)
**OPC-1**	110	34.04–33.03	33.535	17 × 10^−12^
**OPC-2**	110	33.94–33.14	33.54	15 × 10^−12^
**Be10-1**	90	34.12–34.06	21.015	4.75·× 10^−12^
**Be10-2**	90	33.92–33.89	33.91	4.6·× 10^−12^

**Table 4 materials-14-01300-t004:** Concentration of chlorides in the surface of the specimen at the end of the experiment and in the surface of the steel bar.

	Bar in Reference	Bar in Be10
Chloride Surface concentration	1.10	1.64
Chloride threshold (% mass mortar)	0.35	0.13

**Table 5 materials-14-01300-t005:** Phenolphthalein staining in the cements studied to determine carbonation depth.

CEMENT	3 MONTHS	6 MONTHS
INDOORS	OUTDOORS UNSHELTERD	OUTDOORS SHELTERED	INDOORS	OUTDOORS UNSHELTERD	OUTDOORS SHELTERED
OPC	10% Be	OPC	10% Be	OPC	10% Be	OPC	10% Be	OPC	10% Be	OPC	10% Be
**CEM I 42.5R**	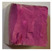	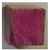	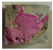	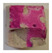	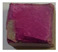	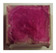	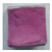	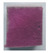	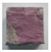	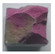	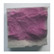	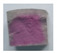
**CEM I 52.5R-SR3**	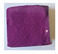	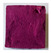	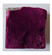	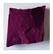	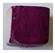	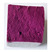	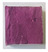	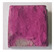	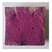	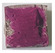	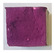	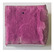
**CEMII/AS42.5N**	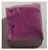	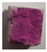	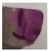	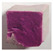	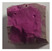	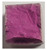	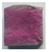	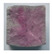	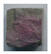	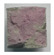	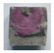	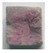
**CEMII/A-P(16) 42.5R**	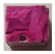	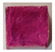	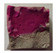	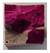	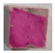	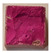	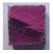	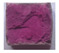	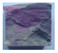	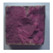	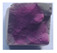	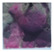
**CEMII/A-P(13) 42.5R**	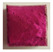	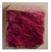	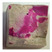	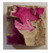	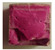	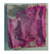	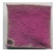	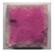	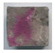	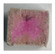	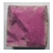	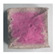
**CEMII/AV42.5R**	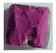	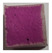	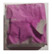	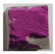	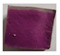	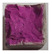	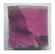	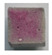	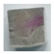	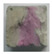	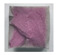	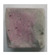
**CEMIII/A42.5N**	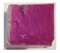	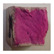	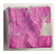	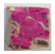	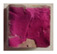	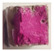	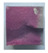	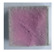	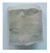	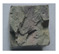	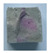	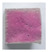
**CEMIV/A(V) 42.5R SR**	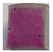	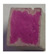	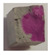	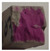	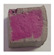	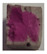	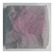	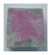	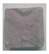	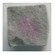	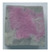	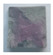
**IV/A(P) 42.5R/MR**	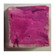	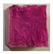	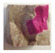	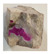	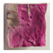	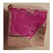	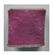	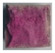	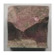	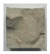	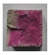	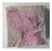
**BLII/B-LL42.5R**	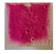	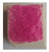	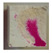	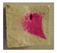	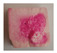	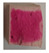	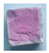	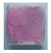	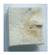	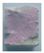	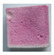	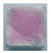

## Data Availability

Data sharing is not applicable to this article.

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
