# Peer review of "Reduced Carbonation, Sulfate and Chloride Ingress Due to the Substitution of Cement by 10% Non-Precalcined Bentonite"

_materials, 2021, doi:10.3390/ma14051300_

Round 1
Reviewer 1 Report
Thank you for submitting this interesting work to materials. This is an interesting and timely study and I hope we can publish this soon. Below are a few comments that could further improve the quality of this work:
General:
-check your affiliations - provide addresses
Abstract:
-try to quantify some of these positive findings - that will make this strong
Introduction:
- I think clinker production contributes to some 5% of all anthoprogenic CO2 emissions - not sure about the exact number, but it is big and you can mention that to show the importance of your work
- This is somewhat biased but in Europe CO2 capture (https://www.mdpi.com/1996-1073/13/21/5692) and solar calcination (https://www.mdpi.com/2227-9717/5/4/67) are very important topics to discuss - maybe you can include this and say why your method is better or how it can complement these methods
- Please provide a reference that we run out of fly ash - I dont really see this happening
- don't call it preliminary research - that means you are not done yet...
Materials and Methods:
- put table 1 in front of fig 1
- Fig. 2 all pictures next to one another and similar height
Results:
- Fig 5 and Fig 6 - put next to one another in one figure and reduce seize
- Fig 9 and 10 make also one figure next to one another
- Do you really need Fig 7 and 8 - elaborate
- what was the rainfall like - quickly mention how the conditions might have differed
- Table 3 - enlarge that over the whole width
Discussion:
- the discussion is OK - really add more numbers - it is a scientific paper so you should present scientific results (numbers) and compare them to other scientific results
- discuss industrial application of this method
Conclusions:
- good - what is your conculsion with regards to applying this on industrial scale - can it be done? can it be done now?
Author Response
Modifications in the new manuscript are marked in red.
Dear Reviewer 1,
Thank you for your valuable comments. We are grateful for their input which has helped to improve the paper.
It is a pleasure to inform you that we have performed the revision according to your suggestions.
Comments and Suggestions for Authors
Abstract:
-try to quantify some of these positive findings - that will make this strong
We have introduced some numbers in the Abstract
Introduction:
- I think clinker production contributes to some 5% of all anthoprogenic CO2 emissions - not sure about the exact number, but it is big and you can mention that to show the importance of your work
Thanks for your valuable comment. We have added the following phrase:
“Nowadays, Portland cement industry is considered as ultimately responsible for about 7.4% of the worldwide carbon dioxide emissions, which accounted for nearly 2.9 Gtons in 2016 [reference A1].”
Reference A1: Sanjuán, M.Á.; Andrade, C.; Mora, P.; Zaragoza, A. Carbon Dioxide Uptake by Cement-Based Materials: A Spanish Case Study. Appl. Sci. 2020, 10, 339. https://doi.org/10.3390/app10010339
- This is somewhat biased but in Europe CO2 capture (https://www.mdpi.com/1996-1073/13/21/5692) and solar calcination (https://www.mdpi.com/2227-9717/5/4/67) are very important topics to discuss - maybe you can include this and say why your method is better or how it can complement these methods
We agree with the reviewer 1: CO2 capture and solar calcination are two highly topical subjects. Therefore, we have included the following text:
“In order to achieve zero carbon dioxide emissions in 2050, the Portland cement industry is assessing the potential measures to be implemented. They are related to the production of clinker, cement, and concrete, as well as to construction measures and carbonation of cement-based materials during their service-life and end-of-use phases [reference A2]”. Currently, it is believed that the existing mitigation technologies are insufficient to achieve the net zero carbon target. Accordingly, innovative technologies such as carbon dioxide capture, utilization, and storage (CCUS) technologies [reference A3] or flameless systems for calcination of Minerals should be implemented by 2050. The tube-in-tube helical system for the calcination of minerals is a new flameless system which use is foreseen in concentrated solar power plants [reference A4].
Reference A2: Sanjuán, M.A.; Argiz, C.; Mora, P.; Zaragoza, A. Carbon Dioxide Uptake in the Roadmap 2050 of the Spanish Cement Industry. Energies 2020, 13, 3452. https://doi.org/10.3390/en13133452
Reference A3: Plaza, M.G.; Martínez, S.; Rubiera, F. CO2 Capture, Use, and Storage in the Cement Industry: State of the Art and Expectations. Energies 2020, 13, 5692. https://doi.org/10.3390/en13215692
Reference A4: Haneklaus, N.; Zheng, Y.; Allelein, H.-J. Stop Smoking—Tube-In-Tube Helical System for Flameless Calcination of Minerals. Processes 2017, 5, 67. https://doi.org/10.3390/pr5040067
- Please provide a reference that we run out of fly ash - I dont really see this happening
Thank for your valuable suggestion. The paper has been modified as follows:
“Energy consumption in Spain by coal power plants decreased from 20.2% in 1990 to 9.8% in 2017 as result of the increase use of alternative energies [reference A5]. In the world, coal represented the 38.5% of the power generation in 2018. However, concerns about greenhouse gas emissions cloud the future of coal because it is at the centre of debate on energy and climate policy. Several countries have declared targets to reach net-zero GHG emissions by 2050 and, consequently, have decided to end coal power generation. Nevertheless, in other countries coal generation plays a key role to accessing affordable energy [reference A6]. In any case, the coal share in global power mix is expected to be reduced to 10% by 2050. Consequently, the availability of coal fly ash in some countries would be reduced.
Reference A5: Secretary of State for Energy of the Ministry for the Ecological Transition (2019) Energy in Spain 2017. National State Administration publications, Madrid, Spain (in Spanish) https://energia.gob.es/balances/Balances/LibrosEnergia/Libro-Energia-2017.pdf
Reference A6: International Energy Agency, IEA (2018) Coal 2018. IEA Publications. Paris, France. https://www.iea.org/reports/coal-2018
- don't call it preliminary research - that means you are not done yet...
Thanks, we have removed
Materials and Methods:
- put table 1 in front of fig 1
The figure has been placed before the table
- Fig. 2 all pictures next to one another and similar height
Made
Results:
- Fig 5 and Fig 6 - put next to one another in one figure and reduce seize
- Fig 9 and 10 make also one figure next to one another
- Do you really need Fig 7 and 8 - elaborate
- what was the rainfall like - quickly mention how the conditions might have differed
- Table 3 - enlarge that over the whole width
All changes made
Discussion:
- the discussion is OK - really add more numbers - it is a scientific paper so you should present scientific results (numbers) and compare them to other scientific results
We do not understand well the suggestion by the reviewer. We have presented numbers in the results. In any case we have enlarged some paragraphs mentioning the previous results from other authors although in carbonation and chloride resistance we have not found results previous than ours.
- discuss industrial application of this method
The industrial application is very immediate as in the cement plant it is normal to use mineral additions added usually to the grinding mill together with the clinker. The bentonite we have used is already grinded but there are other that are not. If they are a powder already, what is needed is to homogenise the mixing with the cement.
In the concrete mixer it can be added directly as a powder.
Conclusions:
- good - what is your conclusion with regards to applying this on industrial scale - can it be done? can it be done now?
Thank for your valuable suggestion. The conclusion with regard to the use of bentonite on industrial scale is written as follows:
“New and innovative measures must be undertaken by the worldwide cement industry to minimize its impact on the climatic change. An important lever to reach carbon neutrality is the use of new Portland cement constituents. Given that, bentonite is a clay, but it does not need any thermal activation. However, it is suggested that bentonite could be classified as natural calcined pozzolana (Q)” according to the European standard EN 197-1:2011.”

Reviewer 2 Report
Very nicely done and pertinent paper. See below my comments for consideration before the work can be accepted for publication:
- The language is confusing and the paper is very hard to understand. Please read and review carefully and fix errors with language. Please note many, many typos as well and fix these(see L147 as an example with random periods in the text).
- Figure 1: Please describe how this data was obtained. Fineness of the material does not seem much higher than cement so please clarify.
- P4: Was the w/c constant or was the w/cm constant (cm including bentonite). Clarify and justify.
- Section 2.3.1: What was the paste sample preparation?
- I dont follow Table 2, please describe it in the text. Also correct table numbers.
- Please mention CoV values or show errors on the figures.
- Please explain Fig. 8 a bit better.
- Table 3 with the carbonation results is very blurry. The tables are also labelled wrong. Please fix.
- The work needs to be referenced better. Many claims but no references to back up the claims.
Author Response
Modifications in the new manuscript are marked in red.
Dear Reviewer 2,
Thank you for your valuable comments. We are grateful for their input which has helped to improve the paper.
It is a pleasure to inform you that we have performed the revision according to your suggestions.
Very nicely done and pertinent paper.
Thanks for the comment
See below my comments for consideration before the work can be accepted for publication:
- The language is confusing and the paper is very hard to understand. Please read and review carefully and fix errors with language.
The paper has been translated into English by a native. The translator is that employed for the translations in a technical Journal usually made bilingual (Spanish. English) very much experienced with the terminology. The person has been reviewing again the version once incorporating all new comments.
Please note many, many typos as well and fix these(see L147 as an example with random periods in the text).
We have not found in Line 147 any typo. We have tried to find other typos and correct them.
- Figure 1: Please describe how this data was obtained. Fineness of the material does not seem much higher than cement so please clarify.
We have added that the granulometry was made with a laser granulometer and we give the trade mark
We had already in the text that the finesse is smaller than that of cement but not much samller.
- P4: Was the w/c constant or was the w/cm constant (cm including bentonite). Clarify and justify.
The w/c ratios used are now indicated.
- Section 2.3.1: What was the paste sample preparation?
The paste was prepared by adding the bentonite as a powder to the cement and after the water is added. For the XRD sample preparation it is grinding and sieving 45 microns.
- I dont follow Table 2, please describe it in the text. Also correct table numbers.
Thanks it has been corrected
- Please mention CoV values or show errors on the figures.
We have added the number of specimens but we think that as not being too many samples and that this is research the CoV is not possible to be given
- Please explain Fig. 8 a bit better.
Thanks made
- Table 3 with the carbonation results is very blurry. The tables are also labelled wrong. Please fix.
We have corrected and try to present cleaner. As it has many small pictures it is not an easy figure, but we think that it is very illustrative
- The work needs to be referenced better. Many claims but no references to back up the claims.
We do not understand the meaning of your request. We have added some more comments on the previous literature. We hope this is your request

Round 2
Reviewer 1 Report
Well done- you put considerable work in revising this, and you did a good job. This is fine with me.
Author Response
Thank you very much for your comment.
Reviewer 2 Report
Either there is something wrong with your .pdf file or with the conversion, or with my .pdf file. This is how the text is showing for me "As ettringite content. in turn. followed neither an upward nor a downward pattern." There are numerous random periods all through the text. There are still numerous issues with language. I dont have major technical issues but the work is hard, if not impossible to read and follow in its current state.
Author Response
Dear Reviewer 2.
Our sincere apologies. We don't really know why, but in the version of changes, commas and dots have been interchanged in some paragraphs making the article extremely hard to read, as you say.
However, in the final pdf version they were set correctly.
We have updated this version (with changes) also correcting it proppertly.
Related with the comment about Extensive editing of English language and style required, in our opinion, it must also be for this reason. The article has been translated and revised again, in this version by a native professional translator.
Thank you very much for your comment about you don't see major technical issues.